

# Emotion generation method in online physical education teaching based on data mining of teacher-student interactions

Yanwei Zhao[1], Xiangyun Kong[2], Wei Zheng[3] and Shahbaz Ahmad[4]

[1] Langfang Normal University, LangFang, China
[2] Hebei Oriental University, LangFang, China
[3] Langfang 16th Middle School, LangFang, China
[4] National Textile University, Faisalabad, Pakistan

## ABSTRACT

Different from conventional educational paradigms, online education lacks the direct interplay between instructors and learners, particularly in the sphere of virtual physical education. Regrettably, extant research seldom directs its focus toward the intricacies of emotional arousal within the teacher-student course dynamic. The formulation of an emotion generation model exhibits constraints necessitating refinement tailored to distinct educational cohorts, disciplines, and instructional contexts. This study proffers an emotion generation model rooted in data mining of teacher-student course interactions to refine emotional discourse and enhance learning outcomes in the realm of online physical education. This model includes techniques for data preprocessing and augmentation, a multimodal dialogue text emotion recognition model, and a topic-expanding emotional dialogue generation model based on joint decoding. The encoder assimilates the input sentence into a fixed-length vector, culminating in the final state, wherein the vector produced by the context recurrent neural network is conjoined with the preceding word's vector and employed as the decoder's input. Leveraging the long-short-term memory neural network facilitates the modeling of emotional fluctuations across multiple rounds of dialogue, thus fulfilling the mandate of emotion prediction. The evaluation of the model against the DailyDialog dataset demonstrates its superiority over the conventional end-to-end model in terms of loss and confusion values. Achieving an accuracy rate of 84.4%, the model substantiates that embedding emotional cues within dialogues augments response generation. The proposed emotion generation model augments emotional discourse and learning efficacy within online physical education, offering fresh avenues for refining and advancing emotion generation models.

## INTRODUCTION

With the development of online technology, more and more people are using online education platforms for learning. The advantages of online education lie in its temporal and spatial flexibility, cost savings, and ease of information sharing. At the same time,

Corresponding author
Xiangyun Kong,
kongxiangyun66666@163.com

through multimedia technology, online education can also provide students with a more vivid and intuitive learning experience. However, in physical education, students need observation and instruction to master movements and skills, and the limitations of online education platforms can negatively affect teaching effectiveness and students' learning experience (*Karagöz, Dinç & Kaya, 2022*; *Xiong, Liu & Huang, 2022*). Especially in the absence of direct teacher-student interaction, it is difficult for students to receive immediate feedback and guidance, and they are prone to learning anxiety and frustration, which affects students' interest and motivation in learning.

Emotional communication refers to the transmission and reception of emotions, feelings, and affective cues through various online communication channels within the realm of physical education instruction. It encompasses the ability of instructors to convey and perceive emotions, empathy, enthusiasm, and encouragement, as well as students' expressions of their emotional states during online physical education classes. This includes verbal cues, tone of voice, body language (when visible through video), choice of words, and any other means through which emotions are conveyed and understood in the online teaching environment. This study aims to explore an emotion generation model based on data mining of teacher-student course interactions to improve online physical education's teaching quality and learning effectiveness. The affective information is extracted through the course interaction between analysts and students. Then, the corresponding affective feedback is generated to help students master the movements and skills and improve the learning effect (*Li et al., 2022*). At the same time, the model can also provide teachers with timely feedback and guidance to help them optimize their teaching methods and strategies and improve teaching quality and effectiveness (*Huang et al., 2021*; *Li, Ortegas & White, 2023*).

Compared with existing studies, this study proposes an emotion generation model based on data mining of teacher-student course interactions for online physical education, including data preprocessing and data enhancement techniques, a multimodal conversation text emotion recognition model and a topic-expanding emotion conversation generation model based on joint decoding. In terms of data preprocessing and data enhancement techniques, this article preprocesses and visualizes the dataset and uses EDA techniques for data enhancement to build a pre-trained model in sentiment dialogue recognition experiments. In terms of the multimodal conversation text sentiment recognition model, this article significantly improves the accuracy of sentiment extraction by extracting contextual features of sentiment conversation text and classifying mixed data with a random forest algorithm using unstructured conversation information and structured user attributes (*Nie et al., 2023*; *Liu et al., 2023*). Regarding the topic-extended sentiment dialogue generation model based on joint decoding, this article extracts topic words by the Twitter-LDA algorithm. It uses them as additional inputs to make the generated responses contain topics and contents that remain consistent with the inputs. The experimental results show that the sentiment generation model proposed in this study has good performance and effectiveness.

Sentiment generation models for teacher-student course interactions in online education face multifaceted challenges. These challenges include difficulties in grasping

contextual nuances such as subject-specific language and cultural influences, an inability to interpret non-verbal cues prevalent in face-to-face interactions, the subjective nature of emotions in educational settings, the diverse expressions of emotions across individuals and cultures, and the dynamic, real-time nature of interactions. Overcoming these hurdles demands models equipped to comprehend context, decode non-verbal cues, handle subjectivity, accommodate diverse expressions, and flexibly adapt to the dynamic nature of online educational interactions (*Yang, Zhang & Kim, 2022*).

The main contribution of this study is to propose an effective generation model for online physical education to improve teaching quality and learning outcomes.

Development of an affective generation model: Utilizing deep learning and natural language processing techniques, the study proposes an affective generation model. This model leverages course interaction data between analysts and students to extract emotional information. It generates tailored feedback that aids students in improving their mastery of movements and skills, consequently enhancing learning effectiveness. Simultaneously, this model offers teachers timely feedback and guidance to refine their teaching methods and strategies, thereby augmenting teaching quality and effectiveness (*Shutova & Andryushchenko, 2020*).

Exploration of advanced techniques: The study delves into various advanced techniques such as data preprocessing, data enhancement, multimodal dialogue text emotion recognition, and a novel topic-expanding emotion dialogue generation model based on joint decoding. These explorations introduce new methodologies and ideas for implementing emotion generation models in educational settings.

Proposal of an emotion generation model: Through data mining of teacher-student course interactions, the study proposes an emotion generation model. This model extracts emotional features from dialogues between teachers and students to generate emotional feedback, facilitating students' improvement in mastering movements and skills. Additionally, it assists teachers in refining their teaching strategies. The study's contributions offer practical value and significance in the application of affective generation models in education.

Overall, this paragraph effectively summarizes the key contributions of the study, emphasizing its potential impact on improving teaching quality and learning outcomes in online physical education through the development and application of innovative models and techniques.

## RELATED WORK

### Emotional dialogue system

In everyday communication, when two parties talk about a certain topic, they usually express their emotions. Therefore, in a human-computer dialogue system, the machine must have a certain level of emotional computing capability to achieve a long conversation between the machine and the user and be able to perceive changes in emotion. This requires that the machine first identify and judge the emotions implied by the speaker accurately and generate emotionally rich responses. According to existing research, an affective dialogue system can be divided into two main parts: dialogue emotion recognition

and emotion dialogue generation (*Viscione & D'Elia, 2019*). As shown in Fig. 1, it is the framework of an effective dialogue system.

## Conversational text-based sentiment recognition method

Emotions are vital for effective communication in dialogue systems, adding depth to language. Recognizing user emotions is crucial in dialogue systems, as emotions play a significant role in interpersonal dynamics. Identifying emotions in a conversation involves tracking changes in both parties' emotions to infer the speaker's current emotional state. Sentiment recognition has gained attention in natural language processing due to the popularity of the dialogue system.

Researchers have explored various approaches for sentiment recognition in dialogues. *Chen et al. (2017)* studied factors influencing emotions in text, while others achieved state-of-the-art performance by modeling emotional interactions using predicted emotion labels. *Shum, He & Li (2018)* used common-sense knowledge for emotion recognition. Sentiment recognition is crucial for interpreting fine-grained dialogue systems and downstream tasks.

Sentiment recognition in dialogue text involves analyzing sentiment-related features. It can result in a binary or multiclass sentiment classification. Multiclass classification is useful for analyzing sentiment in comments and understanding public opinion trends. Machine learning techniques like neural networks have been widely used in text sentiment analysis. Recent research focuses on neural network structures, attention mechanisms, and combinations. For example, *Caldarini, Jaf & McGarry (2022)* used bidirectional long-short term memory (Bi-LSTM) neural networks for sentiment analysis of Chinese comments, and *Feng & Cheng (2021)* proposed a CNN with word attention mechanisms to learn the hidden sentiment information. LSTM and GRU models have significantly improved sentiment classification accuracy.

However, more research is needed for sentiment recognition in conversational text. Conversational sentiment recognition analyzes sentiments in social media conversations, considering contextual connections between sentences and chronological order for modeling. These systems consist of contextual information sentiment recognition and user information sentiment recognition. LSTM networks extract contextual features and analyze the speaker's implied sentiments. However, associative dependencies between sentences are not considered. Researchers used multi-layered recurrent neural networks to address this for better contextual information modeling, sentiment analysis, and improved accuracy.

Shifts in the speaker's affective state during a conversation have been analyzed using user attribute features. Models incorporating historical dialogue information analyze user emotions. Graph convolutional neural networks combine user information and emotional, conversational text using a graph structure with user features as nodes and utterance dependencies as edges.

Conversational sentiment recognition is applied in various scenarios, including detecting user sentiment changes in human-computer systems, sentiment analysis of user comments in social media, and customer sentiment on e-commerce platforms.

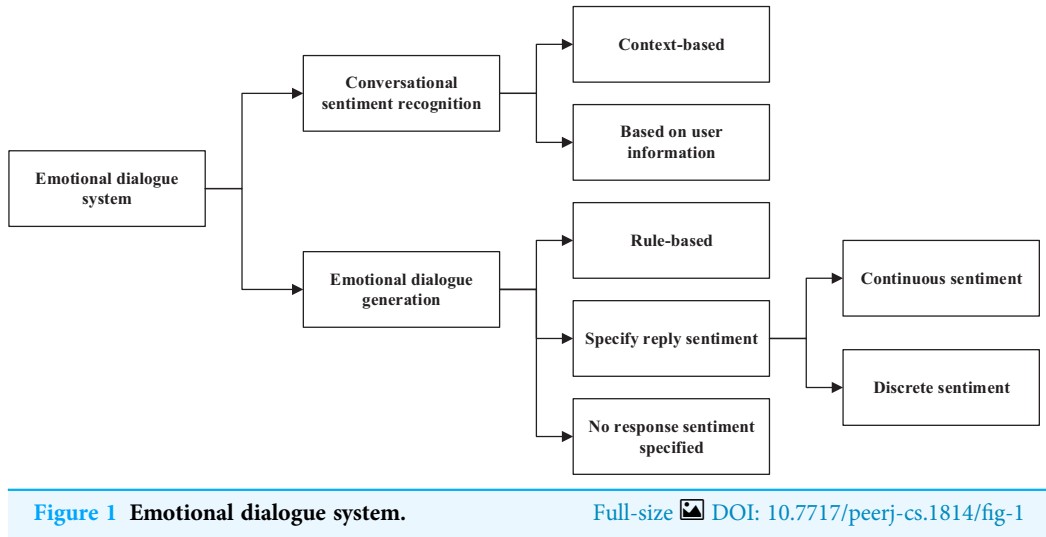

**Figure 1** Emotional dialogue system.               

## Conversational text-based emotional dialogue generation methods

Conversational text-based approaches to affective dialogue generation are highly respected in HCI systems, and researchers are increasingly interested in exploring dialogue systems. These systems are typically categorized as retrieval-based or generation-based. Retrieval-based approaches rely on a large dialogue *corpus* and select responses based on matching scores (*Bishop, 2021*). However, this approach's results are heavily influenced by *corpus* size and building a large database is time-consuming and labor-intensive. On the other hand, generation-based systems solve the sequence mapping problem using an end-to-end sequence generation architecture. They encode input posts into contextual semantic vectors and generate responses using decoders.

Attention mechanisms were introduced to capture speaker dependencies, enhancing reply diversity and richness. Employing topic information extraction and embedding has aimed to refine response quality to mimic human-like conversational nuances. Nonetheless, prior studies primarily emphasized grammatical accuracy, diversity, and topical relevance while sidelining emotional expressiveness. Attempts to rectify this discrepancy involved training generative dialogue models on casual conversational data, yielding human-like but emotionally deficient responses.

Recognizing this shortfall, researchers have redirected efforts toward enhancing conversational quality. Notably, attention mechanisms were leveraged to grasp speaker dependencies, yielding responses of increased richness and diversity. Concurrently, employing topic information extraction and embedding facilitated the generation of personalized responses. Despite advancements in grammatical accuracy, diversity, and topical relevance, generative dialogue models continue to grapple with expressing sentiment effectively.

However, some studies have explored emotional human-machine chat systems using memory network structures. These models generate responses based on different emotional labels by incorporating emotional embedding memory, internal memory for emotional states, and external memory mechanisms. A pre-training-based dialogue

generation method has also been proposed utilizing a large-scale pre-trained Transformer language model. Researchers have improved the GPT-2 model in affective dialogues by incorporating affective class labels as additional input or using affective categories as generation conditions to enhance emotional expression (*Sutskever, Vinyals & Le, 2014*). These research efforts have significant application prospects and commercial value, making sentiment dialogue generation systems a topic of great interest. Constructing dialogue systems that consider sentence relevance, diversity, and rich sentiment is of great practical importance.

## METHOD

### Emotional text generation techniques and analysis

#### *Emotional text generation techniques*

To study how to incorporate emotions in interactive dialogues in physical education, it is first necessary to investigate how to generate texts containing emotions. The field of affective text generation has received less attention, and few effective methods have been proposed (*Chen et al., 2017*). A more representative one is the Affect-LM model proposed in 2017, which can generate affective text with additional parameters to customize the emotional content in the generated text without sacrificing grammatical correctness. The model is based in the long short-term memory neural network introduced in the previous section, and for a sequence of $w_1, w_2, \ldots, w_M$ with M words, the joint probability of all words of the long short-term memory neural network is calculated by Eq. (1).

$$P(w_1, w_2, \ldots, w_M) = \prod_{t=1}^{t=M} P(w_t | w_1, w_2, \ldots, w_{t-1}) \tag{1}$$

If there are a total of V words in the system vocabulary, then the conditional probability of the word at moment t can be calculated by its context $c_{t-1} = P(w_1, w_2, \ldots, w_{t-1})$ and Eq. (2).

$$P(w_t = i, |c_{t-1}) = \frac{\exp(U_i^T f(c_{t-1}) + b_i)}{\sum_{j=1}^{V} \exp(U_j^T f(c_{t-1}) + b_j)} \tag{2}$$

where $f$ represents the long short-term memory neural network that takes contextual information as input to the network after one-hot processing, U is the matrix of visual word representations associated with POS (lexical), and b is the bias term that appears in the one-shot grammar when capturing words. The equation represents the conditional probability of the long- and short-term memory neural network model without using any additional emotional information 1.

The Affect-LM model performs word prediction with an additional energy term added to the long and short-term memory neural network prediction formula, which can be calculated by Eq. (3).

$$P(w_t = i, |c_{t-1}) = \frac{\exp(U_i^T f(c_{t-1}) + b_i) + \beta V_i^T g(e_{t-1}) + b_i}{\sum_{j=1}^{V} \exp(U_j^T f(c_{t-1}) + b_j) + \beta V_j^T g(e_{t-1}) + b_j} \tag{3}$$

where $e_{t-1}$ is the sentiment vector consisting of the sentiment category information, g is

the network that processes the sentiment vector, $V_i$ is the word vector of the first word in the vocabulary, and the parameter B is the sentiment intensity parameter, which represents the degree of influence of context and sentiment information on the target word prediction, and when $\beta = 0$, it means that the text generation is completely unaffected by sentiment. The given equation shows that the function of Affect-LM relative to the long and short-term memory neural network with the extra energy term is to add the affective factor to the normal text generation process.

The study of the Affect-LM model shows that the main problem of generating text with sentiment is how to quantify the sentiment factor for model learning, and the same is true for generating responses with sentiment in dialogue systems (*Shao et al., 2017*).

(1) Mapping sentiment to sentiment vectors

Most models proposed in the field use sentiment vectors to map sentiments. Each sentiment category is assigned a sentiment category vector, denoted as V, incorporated into the model as a sentiment factor. Different researchers have provided various approaches to processing these sentiment vectors. *Ling et al. (2021)* took a different approach by introducing an external cognitive emotion lexicon to enhance traditional word embeddings using a three-dimensional emotion space.

(2) Using an objective function with sentiment

The loss function evaluates the difference between predicted and true values in the dataset, and a well-designed loss function leads to better model performance. Three loss functions with sentiment factors optimize the model's sentiment-related objectives (*Chen et al., 2017*).

Minimizing affective dissonance: The authors assume that sentiment does not rapidly shift during a chat. This loss function considers cross-entropy and measures the similarity between the sentiment of generated responses and input sentences using Euclidean distance. The goal is to ensure emotional consistency.

Maximizing affective dissonance: The authors also assume that receiving overly friendly messages from strangers may trigger negative emotions. In this case, the sentiment of the generated response should be opposite to the sentiment of the input sentence, creating emotional inconsistency.

Maximizing emotional content: The authors aim to generate responses with explicit sentiment characteristics, regardless of whether the sentiment is positive or negative. This loss function prevents the model from generating meaningless responses such as "yes" or "I don't know."

Using sentiment vectors or objective functions with sentiment aims to make the model generate sentiment-infused responses. However, current research in sentiment dialogues mainly focuses on generating responses with specified sentiment in single-round dialogues without considering multi-round dialogues. People choose the appropriate emotion to respond to in daily human conversations, which are typically multi-round and contextual. Models for emotional conversations should be able to achieve this effect. Learning to change emotions based on a single round of dialogue is insufficient due to limited information. Introducing multiple rounds of dialogue enables the model to learn the

patterns of emotion changes and incorporate contextual information, resulting in more human-like responses.

## Emotional conversation model (ECM)

After research, although the field of emotional dialogue has been developed for a relatively short time, several more effective methods and models have been proposed, as described specifically in the previous section. This article is based on the emotional conversation model (ECM) model for improvement, and the ECM model (*Xing et al., 2017*) is briefly introduced in this section.

The ECM model is the first complete model proposed in the field of emotional dialogue, which is based on an end-to-end approach where both the encoder and decoder consist of a gated recurrent neural network and two sentiment-related modules are added to the base model to add sentiment factors to the responses generated by the model. Figure 2 shows the four modules: sentiment classifier, sentiment embedding, internal memory, and external memory.

### Sentiment classifier

In the study of sentiment dialogues, obtaining the dataset is difficult. Hence, the authors used a sentiment classifier to label the sentiment of a single round of short textual dialogue datasets. The authors chose the best bidirectional long and short-term memory neural network for sentiment classification as the sentiment classifier; however, according to the previous experimental results, the accuracy of bidirectional long and short-term memory neural network classification is only 0.623. Although this method can construct a dialogue dataset containing sentiment labels, about one-third of the sentiment classification of sentences is wrong. Moreover, the dataset used in the ECM model is a single-round dialogue dataset, which does not apply to the multi-round dialogue studied in this article.

### Sentiment embedding module

Sentiment category is a rather abstract concept, so for each sentiment category, a low-dimensional vector is used to represent it. The sentiment embedding module's function is to initialize a sentiment vector for each sentiment category randomly. The sentiment vector is continuously trained during the training process, and finally, the state $s_t$ of the GRU decoder is updated during the decoding process based on the sentiment vector and the word vector $e(y_{t-1})$, and the update formula is Eq. (4).

$$s_t = GRU(s_{t-1}, [e(y_{t-1}); v_e]) \tag{4}$$

If the sentiment vector is used directly as input, the sentiment category vector does not change during the generation process, which may compromise the grammatical correctness of the sentence. The authors of the ECM model propose using an internal memory module to represent the sentiment dynamics during the decoding process. The authors argue that human emotion in conversation is a gradual process of decline; at the beginning of the sentence, the emotion is very strong, but in the process of sentence generation along with the expression of emotion, the amount of remaining emotion to be

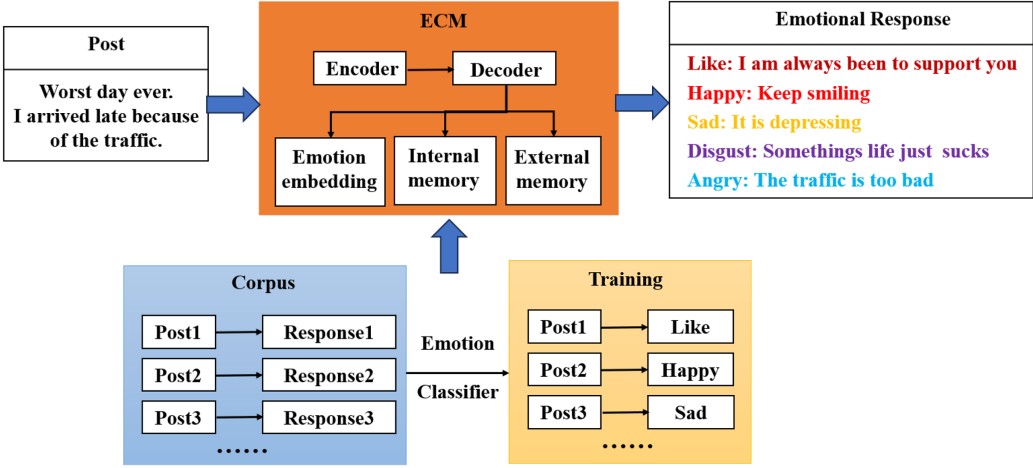

**Figure 2 Basic structure of the ECM model.**

expressed is decreasing. Hence, the intensity of emotion is weaker than at the beginning, when the sentence generation ends; the expression of emotion should also end. The structure of the internal memory module is shown in Fig. 3.

In the decoding process, for each word of each sentence generated, there is a current sentiment state $M_{e,t}^I$ equivalent to the sentiment vector proposed in the previous section. On the other hand, the internal memory module controls how to update the sentiment state in the next step when it is ready to generate the responding word, and this update is done through the write gate and the read gate.

The write gate updates the sentiment state before it enters the gated recurrent neural network. The write gate is computed concerning two variables: the word vector $e(y_{t-1})$ of the last input word during decoding and the last state $s_{t-1}$ of the decoder, and the write gate is computed with Eq. (5).

$$g_t^r = sigmoid(W_t^r[e(y_{t-1}); s_{t-1}]). \tag{5}$$

The read gate is only related to the current state of the encoder, and the values of the read gate during the encoding process are all less than 1. Therefore, in each update step, the sentiment state is gradually decayed. When the sentence generation is over, the sentiment state can be decayed to 0 so that all the sentiment is fully expressed, and the formula to calculate the read gate is as in Eq. (6).

$$g_t^w = sigmoid(W_g^w s_t). \tag{6}$$

Both the read and write gates are used to update the sentiment state of the model. After the activation function processing, the values of both the read and write gates are located in the (0, 1) interval, and the sentiment state is an ordinary sentiment vector without any other information when the write gate does not process it. Since the write gate and the word vector of the previous input word, the previous state of the decoder, and the context vector are related, the sentiment state can be dynamically associated with these variables

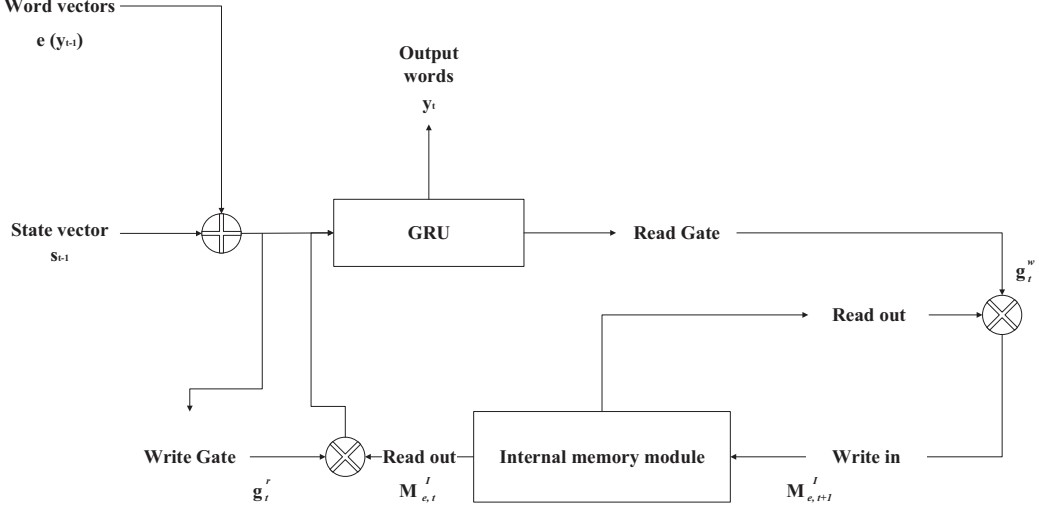

**Figure 3 Structure diagram of internal memory module.**

after the write gate processing. The read gate then controls the relationship between the initial value of the next and current sentiment states. Since the value of the read gate is always less than one during the decoding process, the initial value of the next sentiment state will always be smaller than the current sentiment state and infinitely close to 0 when the sentence reaches the end. The sentiment state is updated by Eqs. (7) and (8).

$$M_{r,t}^I = g_t^r \otimes M_{e,t}^I \tag{7}$$
$$M_{e,t+1}^I = g_t^w \otimes M_{e,t}^I \tag{8}$$

After obtaining the sentiment state processed by writing gates, the gated recurrent neural network model updates the current moment state of the decoder using the newly obtained sentiment state instead of the sentiment vector, and the update is calculated by Eq. (9).

$$s_t = GRU(s_{t-1}, [e(y_{t-1}); M_{e,t}^I]). \tag{9}$$

### External memory module

The external memory module's purpose is to control the model's expression by assigning different generation probabilities to sentiment and generic words. Although a sentence filled with emotional words can express the corresponding emotion, it will inevitably affect the sentence's fluency in grammatical expression. Hence, emotion and the generation probabilities of generic words need to be controlled. A controls the probability of generating generic words and emotion words. When the next word is generated in the decoding stage of the model, there is a probability of taking a word from the emotion word list and a 1-a probability of taking a word from the generic word list. A word from the generic word list, as shown in Fig. 4, is the structure of the external memory module. The

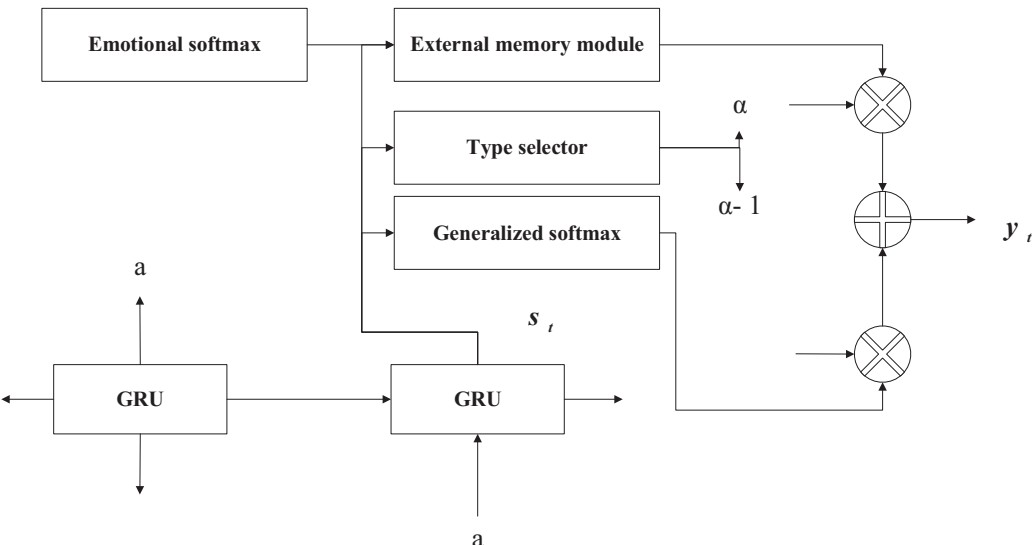

**Figure 4 External memory module structure diagram.**

external memory module can find a balance between emotional expression and grammatical fluency.

The ECM model has some problems; firstly, when choosing the dataset as the basis of the experiment, the authors used a two-way long- and short-term memory model because there is no short conversation set with sentiment classification. The neural network sentiment classification model was used to label the sentiment of the existing short conversation set (NLPCC). Although the two-way long and short-term memory neural network is the best model for sentiment classification, its accuracy is only about 0.623, which means that one-third of the data in the dataset is incorrectly classified. As the training base of the model, its errors can greatly affect the model effect and conclusion. Through article reading and data query, we found that there exist dialogue datasets with sentiment markers in the field of dialogue sentiment classification, such as IEMOCAP (*Gao, Galley & Li, 2018*), SEMAINE (*Zhang et al., 2019*), Emotionlines (*Hu et al., 2017*), MELD (*Huang et al., 2018*) DailyDialog (*Zhou et al., 2018*) and EmoContextl (*Asghar et al., 2018*). The IEMOCAP, Emotionlines, and MELD datasets are multimodal (containing sound, visual, and textual information), and the DailyDialog and EmoContext datasets are text-based. The distribution of sentiment labels for these datasets is shown in Table 1, and DailvDialog is chosen as the dataset for the thesis model in this thesis.

Secondly, the research goal of the ECM model is a single-round dialogue. The sentiment information contained in a single-round dialogue conversation is limited. The model cannot learn the sentiment change pattern during the dialogue, so the ECM model can only specify the model to conduct the dialogue with a particular sentiment. However, the ultimate goal of the emotional dialogue model should be for the model to learn the emotional changes during the dialogue and determine which emotion should be used to reply through learning instead of manually specifying which emotion the model should use to reply.

**Table 1 Distribution of sentiment labels in common sentiment dialogue datasets.**

| Dataset | DailyDialog | MELD | EmotionLines | IEMOCAPEmoContext | Total |
|---|---|---|---|---|---|
| Neutral | 85,572 | 6,436 | 6,530 | 1,708 | 101,246 |
| Happy | 12,885 | 2,308 | 1,710 | 648 | 18,651 |
| Surprised | 1,823 | 1,636 | 1,658 | 1,084 | 6,201 |
| Sadness | 1,150 | 1,002 | 498 | 1,103 | 2,753 |
| Anger | 1,022 | 1,607 | 772 | 4,669 | 8,070 |
| Disgusted | 353 | 361 | 338 | 5,838 | 5,890 |
| Fear | 74 | 358 | 255 | 5,954 | 6,641 |
| Frustration | N/A | N/A | N/A | 1,849 | 1,849 |
| Other | N/A | N/A | N/A | 1,041 | 1,041 |

To address the second problem of ECM models, this article proposes a method for modeling emotions in multi-round dialogues; by modeling emotion changes, the model can learn the pattern of emotion changes in human dialogues and which specific emotion category should be used to answer for a certain emotion input (*Sun et al., 2018*). We use a long-and short-term memory neural network to model the change of emotion in multi-round conversations to accomplish the task of emotion prediction.

## Multi-turn dialogue generation techniques and analysis

Contextual information plays a pivotal role in sentiment generation as it provides the necessary background for the model to understand and interpret the sentiment correctly. This contextual information encompasses various elements such as the surrounding words, phrases, tone, domain-specific knowledge, cultural nuances, and the overall context of the conversation or text.

Despite being relatively new, researchers have proposed methods that leverage contextual information in the field of multi-turn dialogue. These models can be broadly categorized into two types: non-hierarchical models and hierarchical models. Non-hierarchical models directly combine the context sentences with the input sentence as the model input. The selection of specific context sentences to concatenate depends on the desired effect. On the other hand, hierarchical models utilize utterance-level models to capture each sentence's meaning and integrate context and input information for processing. Non-hierarchical models significantly increase the input length by concatenating context and input. Although recurrent neural networks can handle long input sequences, this approach risks overlooking crucial information. Generally, hierarchical models outperform non-hierarchical models.

Hierarchical models employ various methods for processing the context information vector. These methods include addition, concatenation, training a recurrent neural network on the context and input information, and weighted addition and concatenation. Figure 5 illustrates commonly used techniques for extracting contextual information in hierarchical models, including word vector addition, word vector concatenation, encoding

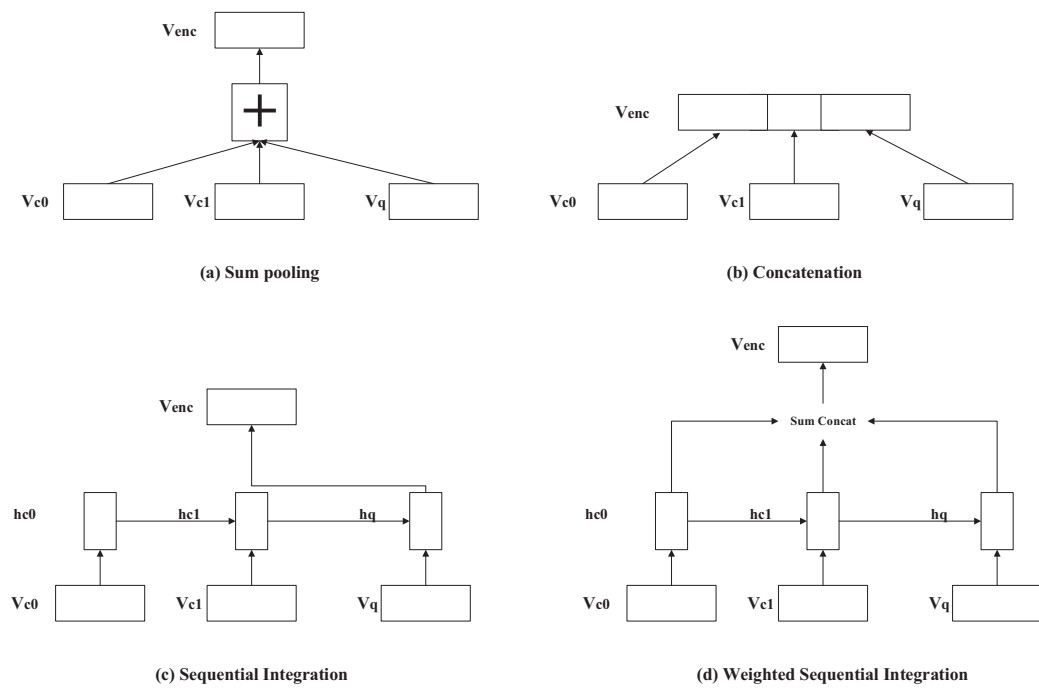

**Figure 5 (A–D) Several ways of handling contextual information in the hierarchical model.**

the context vector using a model, and generating a context vector by adding or concatenating hidden vectors with varying weights at different time steps.

Taking hierarchical models as an example, researchers in multi-turn dialogue have proposed new model structures to utilize contextual information. *Song et al. (2019)* introduced a novel hierarchical recurrent neural network structure called HRED. HRED adds an encoder to the traditional end-to-end model, allowing context modeling and reducing the computational steps between adjacent sentences. This facilitates information transfer in the encoding and decoding processes for multi-turn dialogue. The HRED model consists of the following two modules:

Encoder recurrent neural network: The encoder in HRED is the same as the one in a regular end-to-end model, which encodes the input sentences into fixed-length vectors as final states.

Context recurrent neural network: For n sentences in a multi-turn dialogue, the encoder encodes each sentence, resulting in n sentence vectors. These n-sentence vectors serve as inputs to the context recurrent neural network encoder at different time steps, ultimately generating a fixed-dimensional context vector. Since the context-recurrent neural network module encodes all n sentences, the generated vector contains more comprehensive semantic information.

Decoder recurrent neural network: The vector generated by the encoder recurrent neural network serves as the initial state of the decoder. The vector generated by the context recurrent neural network is concatenated with the word vector of the previous word and serves as the input to the decoder.

# EXPERIMENTS AND ANALYSIS

## Training setting

The running environment of stests included:

Hardware specifications: The experiments were conducted on a server equipped with an Intel Xeon processor (2.5 GHz, 24 cores) and 128 GB of RAM. The computational resources were essential for training and testing the sentiment generation models efficiently.

Software and frameworks: The sentiment generation models were implemented using the Python programming language (version 3.8) and various libraries and frameworks. The primary libraries included TensorFlow (version 2.5) and Keras (version 2.4) for deep learning model development. Additionally, NLTK (Natural Language Toolkit) and scikit-learn (version 0.24) were used for data preprocessing and feature extraction. The parameter settings for model training are shown in Table 2.

## Datasets and training

This article employs the DailyDialog dataset as our research's primary dataset, specifically designed for multi-turn everyday conversations. Compared to previous datasets, this dataset is relatively noise-free and covers many significant topics related to daily life. It provides emotional annotations for each sentence in the dialogues, which align perfectly with the requirements of our model. The DailyDialog dataset comprises 13,000 dialogues, with an average length of eight turns per dialogue.

The initial focus of this article lies in validating the effectiveness of the ECM model, which plays a crucial role in integrating emotions into the generated responses within our proposed system. We replicated the model's results to achieve this, considering that the original ECM model was trained on a single-turn dialogue dataset. We used an open-source English Twitter dialogue dataset to perform the validation, as shown in Fig. 6. This dataset consists of a training set, a target set, and corresponding emotion labels. Since the dataset lacked emotion labels, we incorporated an emotion classification module to annotate the dataset with emotions, ensuring that our ECM model could be effectively validated.

The comprehensive utilization of the DailyDialog dataset, coupled with the integration of the emotion classification module, enhances the robustness of our research and contributes to a more accurate assessment of the effectiveness of the ECM model. This approach allows us to further refine and strengthen the proposed emotion generation method in the context of online physical education teaching (Roy & Das, 2022).

The reproduction of the ECM model involves three key components: the emotion classification module, the preprocessing module, and the ECM model itself. The emotion classification module employs a bidirectional long-short-term memory neural network to accurately assign emotions to the dataset. On the other hand, the preprocessing module is responsible for converting the dataset's words into corresponding IDs (Gamazo & Martínez-Abad, 2020), ensuring compatibility for further training. Finally, the ECM model

**Table 2 Parameter settings for model training.**

| Parameter | Value |
| --- | --- |
| Hidden units | 128 |
| Learning rate | 0.001 |
| Batch size | 64 |
| Dropout rate | 0.4 |
| Regularization parameter | 0.0005 |

undergoes comprehensive training to obtain the final dialogue model, seamlessly incorporating integrated emotions.

We meticulously selected 1,000 data instances during the reproduction experiments as the test set to evaluate the model's performance. We continuously monitored the perplexity of the model on this test set during the training process to assess its capability to generate responses. The results of the reproduction experiments showed impressive outcomes, as after training for 10,000 steps, the loss converged to an exceptional 0.75, and the perplexity on the test set reached a remarkable 1.9. Furthermore, we comprehensively compared the training results of the ECM model and a regular end-to-end structure. Notably, the end-to-end model exhibited a loss convergence of 1.65 and a test set perplexity convergence of 3.4. As depicted in Figs. 7A and 7B, line graphs were plotted to illustrate the specific performance differences between these two models visually.

These remarkable results vividly demonstrate that the ECM model outperforms the end-to-end model in terms of loss convergence and perplexity on the test set, clearly showcasing the superiority of our proposed approach in generating and effectively integrating emotions into responses. The successful reproduction of the ECM model not only bolsters the credibility of our research but also lays the foundation for further advancements in emotion generation within the context of online physical education teaching.

The comparison of the line graphs reveals that both models begin to converge at 10,000 training steps. The loss value of the ECM model converges to 0.7, while the end-to-end model converges to 1.3. On the test set, the trapped value of the ECM model converges to 1.7, whereas the confusion value of the end-to-end model converges to 3.4. Regardless of the loss value or confusion value of the model training, the ECM model is better than the traditional end-to-end model, which proves that the sentiment information in the conversation is helpful for the model to generate responses.

## Sentiment modeling effectiveness

The following are the comparison results of a single-layer long and short-term memory neural network model built using TensorFlow on the DailyDialog dataset, with evaluation metrics including mean absolute error, loss value, and accuracy (*Wang, 2021*). We experimented with different values for word vector dimension, hidden vector dimension, and batch size.

Teacher: Good morning, students! Today we are going to start our physical education class. I am your physical education teacher, and my name is Mary, the teacher. Today we will learn about the basic skills of basketball. Who is interested in basketball?

Student 1: I'm interested in basketball, teacher!

Teacher: Great! Basketball is a very interesting sport. First, let's start with the basic dribbling. Put the ball in your hands and then control the roll of the ball with your fingertips. Try dribbling the ball back and forth across the court, making sure it doesn't roll out of your control.

Student 2: Teacher, I find it difficult to dribble the ball. I often have trouble controlling the ball.

Teacher: It's okay, Student 2. It takes some practice and patience to dribble the ball. You can spend more time practicing and gradually improve your ball handling skills. Also, keep your body in a low position to control the ball better, try it.

Student 3: Teacher, I have a question. In a game, I sometimes get robbed by the other team, is there any way to avoid being robbed?

Teacher: Good question, Student 3. One way to avoid being snapped is to learn to keep your balance and improve your passing skills. When you are being pressed, try to stay calm and watch your teammates on the field for passing opportunities. Don't forget to use your eyes to observe your surroundings, anticipate your opponent's movements, and try to avoid over-dribbling.

Student 4: Teacher, are there any basketball training exercises you can recommend for us?

Teacher: Sure! We can try some basic exercises such as shooting exercises, passing exercises and teamwork exercises. These exercises will help you improve your basketball skills and teamwork. We will learn and practice these exercises step by step in the next class.

Translated with www.DeepL.com/Translator (free version)

**Figure 6 Open source English Twitter conversation dataset.**

As shown in Table 3, the model's optimal sentiment change modeling occurs when using a word vector dimension of 64, a hidden vector dimension of 256, and a batch size of 64. At these parameter values, the model achieves an accuracy of 84.4%.

Once the parameters are established, this article's model assigns a sentiment category to each user-input sentence. This sentiment category corresponds to the sentiment category of the model-generated response. We utilize the ECM model to generate responses based on the user's input sentences and the associated sentiment categories.

## Evaluation of the final effect

After training the model, we proceed to test it. During the testing phase, the encoder encodes the user input, and the resulting encoding is used as input for the decoder. The decoder continues decoding until a complete response is generated (*Shaukat et al., 2016*). Both the trained sentiment prediction model and the contextual information model are involved throughout the decoding process. The sentiment prediction model generates predicted sentiment categories for the decoder, while the contextual information model encodes the sentences and provides contextual information vectors to the decoder. To evaluate the model's performance in providing clear responses to user input, we feed sentences to the model and observe the generated responses (*Karthikeyan, Thangaraj & Karthik, 2020*). Additionally, we output the sentiment prediction results on the interface to verify the proper functioning of the sentiment prediction module. Table 4 shows an example of a multi-round conversation between a user and the model in this article.

It can be seen that the model can output responses corresponding to the user input normally, and the sentiment prediction results with the sentences output in the dialogue interface. The sentiment categories of the model-generated responses align with those predicted by the sentiment prediction model, except for the third sentence. In this case, it

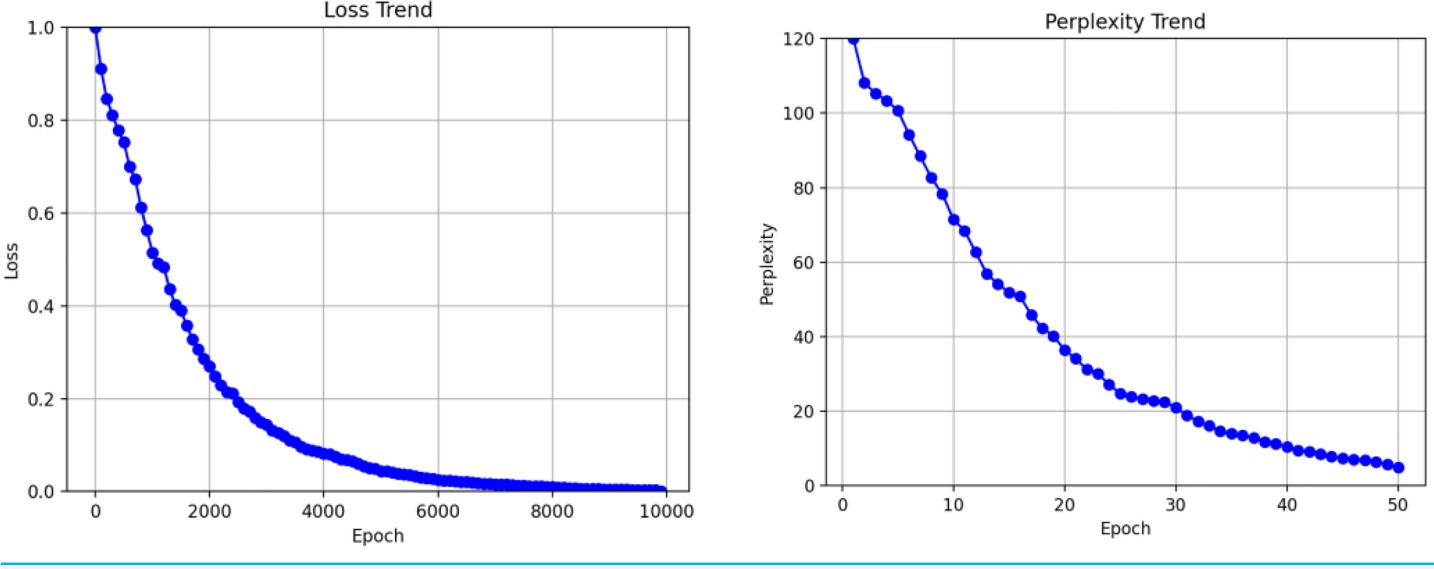

**Figure 7 Training model results.**

leans closer to sadness than no sentiment. However, all other sentiment categories are predicted more accurately.

We have chosen perplexity and accuracy to evaluate the model's performance. Perplexity is used to assess the syntactic effectiveness of sentences, while accuracy is used to evaluate the consistency between the generated sentence's emotion category and the target emotion category. Accuracy is also used to evaluate the accuracy of the generated answer's emotion category. We tested the model in the DailyDialog dataset's test set. We compared the generated answers using our model and an end-to-end model against their actual emotion categories using a bidirectional LSTM-based sentiment classifier. The emotion classification results for the end-to-end model and our model are shown in Table 5.

In addition to that, we conducted a comprehensive comparison between the end-to-end model, categorical emotion states (CES) (*Karthikeyan, Thangaraj & Karthik, 2020*), valence arousal dominance (VAD), and our proposed model. We evaluated the perplexity of the generated results on the test set to assess their performance. The sentiment classification results and the perplexity calculation of the generated sentences were used to create Table 6, which compares the final effects achieved by these four models (*Kragel & LaBar, 2015*).

The findings of our study underscore the notable superiority of our model over the end-to-end CES and VAD models, particularly in terms of perplexity and accuracy. This robust performance attests to our model's exceptional capabilities in generating sentiment-infused responses while ensuring sentence fluency, surpassing the benchmarks set by the end-to-end models. Particularly striking is our model's substantial enhancement in emotional accuracy when crafting responses, a marked improvement compared to the perplexity index. This distinct progress emphasizes our model's significantly bolstered

**Table 3 The effect of model with different parameters fine.**

| Word vector dimension | Hidden vector dimension | Batch size | Mean absolute error | Loss | Accuracy |
|---|---|---|---|---|---|
| 64 | 64 | 32 | 0.0875 | 0.799 | 0.832 |
| 64 | 64 | 64 | 0.0747 | 0.687 | 0.837 |
| 64 | 64 | 128 | 0.0776 | 0.635 | 0.83 |
| 64 | 128 | 32 | 0.099 | 0.733 | 0.807 |
| 64 | 128 | 64 | 0.0984 | 0.695 | 0.837 |
| 64 | 128 | 128 | 0.0813 | 0.697 | 0.823 |
| 64 | 256 | 32 | 0.0759 | 0.748 | 0.836 |
| 64 | 256 | 64 | 0.072 | 0.699 | 0.844 |
| 64 | 256 | 128 | 0.0871 | 0.66 | 0.823 |
| 128 | 64 | 32 | 0.0856 | 0.794 | 0.833 |
| 128 | 64 | 64 | 0.0871 | 0.784 | 0.832 |
| 128 | 64 | 128 | 0.0849 | 0.716 | 0.837 |
| 128 | 128 | 32 | 0.0715 | 0.826 | 0.839 |
| 128 | 128 | 64 | 0.0901 | 0.793 | 0.822 |
| 128 | 128 | 128 | 0.0864 | 0.733 | 0.829 |
| 128 | 256 | 32 | 0.0729 | 0.805 | 0.829 |
| 128 | 256 | 64 | 0.0784 | 0.756 | 0.82 |
| 128 | 256 | 128 | 0.0972 | 0.74 | 0.838 |
| 256 | 64 | 32 | 0.0773 | 0.877 | 0.817 |
| 256 | 64 | 64 | 0.0951 | 0.851 | 0.828 |
| 256 | 64 | 128 | 0.0862 | 0.784 | 0.826 |
| 256 | 128 | 32 | 0.0745 | 0.802 | 0.821 |
| 256 | 128 | 64 | 0.0935 | 0.835 | 0.827 |
| 256 | 128 | 128 | 0.0961 | 0.839 | 0.823 |
| 256 | 256 | 32 | 0.0731 | 0.825 | 0.817 |
| 256 | 256 | 64 | 0.0905 | 0.81 | 0.819 |
| 256 | 256 | 128 | 0.0933 | 0.771 | 0.818 |

**Table 4 Conversation content.**

| Input | Answer | Type |
|---|---|---|
| You didn't ring me last night. | I am sorry to make you disappointed. | Sadness |
| And why were you so rude to me just now. | I didn't mean to be. I do apologize. | Sadness |
| Are you feeling bored? | I'm sorry. I am tired. | No emotion |
| You should have a rest. | Thank you, I will. | No emotion |

capacity to generate emotionally resonant answers, a noteworthy stride highlighted in this research.

These compelling outcomes serve as a testament to the superior performance and efficacy of our model, specifically in the domain of sentiment generation. The

**Table 5 Model sentiment classification results.**

| | Ours | | End-to-end | |
| --- | --- | --- | --- | --- |
| Emotional category | Same | Different | Same | Different |
| Emotionless | 4,160 | 2,161 | 115 | 5,206 |
| Disgust | 5 | 33 | – | – |
| Fear | 10 | 10 | 12 | – |
| Hapiness | 534 | 486 | 256 | 764 |
| Sad | 41 | 43 | – | – |
| Surprise | 57 | 59 | 29 | 87 |
| Total | 4,909 | 2,834 | 1,486 | 6,257 |

**Table 6 Comparing the results of different models.**

| Model | Perplexity | Accuracy |
| --- | --- | --- |
| Ours | 0.76 | 0.92 |
| End-to-end | 0.71 | 0.85 |
| CES | 0.65 | 0.81 |
| VAD | 0.73 | 0.89 |

demonstrated potential of our model to produce emotionally enriched responses in dialogues signifies its pivotal role in advancing natural language processing. By excelling in both accuracy and emotional resonance, our model not only sets a new standard in sentiment generation but also lays a solid foundation for future advancements in generating nuanced and emotionally attuned responses within conversational contexts.

# CONCLUSION

This study contributes a sentiment generation model tailored for teacher-student course interactions. It encompasses data preprocessing, multimodal sentiment recognition, and innovative sentiment dialogue generation techniques. The primary goal was to address challenges in sentiment generation models and propose novel solutions, introducing the ECM model and leveraging the LSTM architecture to capture nuanced sentiment changes during conversations. The LSTM-based model, integrating contextual information with word vectors and sentiment states, demonstrated superior performance in generating heartfelt responses. Through rigorous benchmarking against existing end-to-end models, our approach showcased enhanced sentiment generation capabilities in the context of teacher-student interactions, providing fresh insights into emotional communication and learning dynamics.

While our model exhibits promising results, there are still areas for improvement. Further optimization in architecture and parameter settings could potentially enhance the model's ability to capture subtle sentiment nuances within educational dialogues. Additionally, ethical considerations regarding data privacy and the responsible use of

sentiment analysis in educational settings require careful attention in future iterations. Future studies can focus on refining and optimizing the architecture of sentiment generation models tailored for teacher-student interactions. Exploring variations in LSTM or other recurrent neural network architectures, attention mechanisms, or transformer-based models may enhance the model's ability to capture nuanced sentiment changes more effectively.

### Funding
The authors received no funding for this work.

### Competing Interests
The authors declare that they have no competing interests.

### Author Contributions
- Yanwei Zhao conceived and designed the experiments, prepared figures and/or tables, and approved the final draft.
- Xiangyun Kong conceived and designed the experiments, performed the experiments, performed the computation work, prepared figures and/or tables, and approved the final draft.
- Wei Zheng performed the experiments, authored or reviewed drafts of the article, and approved the final draft.
- Shahbaz Ahmad analyzed the data, authored or reviewed drafts of the article, and approved the final draft.

### Data Availability
The code is available in the Supplemental File.

The data are available at Zenodo: Twitter Dataset. (2022). Twitter Dataset [Data set]. Zenodo. https://doi.org/10.5281/zenodo.7139621.

### Supplemental Information
Supplemental information for this article can be found online at http://dx.doi.org/10.7717/peerj-cs.1814#supplemental-information.

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
