# Peer review of "Emotion generation method in online physical education teaching based on data mining of teacher-student interactions"

_PeerJ Computer Science, doi:10.7717/peerj-cs.1814_

## Round 0.1 · original submission · Major Revisions

Dear authors,

Your article has not been recommended for publication in its current form. However, we encourage you to address the concerns and criticisms of the reviewers, particularly in terms of readability, validity of the findings, and general quality, and to resubmit your article once you have updated it accordingly.

The following points should also be addressed:

1. The abstract should include a statement of the problem you are trying to solve and the purpose of your research, the methods used to find the solution, the results and the implications of your findings. "This study aims to explore an emotion generation model based on teacher-student interaction data mining to improve the teaching quality and learning effects of online physical education teaching" and "This study proposes an emotion generation model based on teacher-student course interaction data mining to optimize emotional communication and learning effects in online physical education teaching" mean the same thing and it should not be repeated.

2. The research gaps and contributions should be clearly summarized in the introduction section. Please evaluate how your study is different from others in the related work section.

3. Please include future research directions.

4. All of the values of the algorithm parameters selected for comparison should be listed.

5. The paper lacks the running environment, including software and hardware. The analysis and configurations of experiments should be presented in detail for reproducibility. It is convenient for other researchers to redo your experiments and this makes your work easy to accept.

6. The authors should clarify the pros and cons of the methods. What are the limitation(s) methodology(ies) adopted in this work? Please indicate practical advantages, and discuss research limitations.
7. Some more recommendations and conclusions should be discussed about the paper considering the experimental results. The conclusion section needs significant revisions. It should briefly describe the findings of the study and some more directions for further research. The authors should describe academic implications, major findings, shortcomings, and directions for future research in the conclusion section. The conclusion in its current form is confused in general. It is strongly suggested to include future research in this manuscript. What will happen next? What are we supposed to expect from the future papers? So rewrite it and consider the following comments:
- Highlight your analysis and reflect only the important points for the whole paper.
- Mention the benefits
- Mention the implication in the last of this section.

8. The values for the parameters of the algorithms selected for comparison should be given.

9. Organization of the paper should be given at the end of the Introduction section.

**Language Note:** The review process has identified that the English language must be improved. PeerJ can provide language editing services - please contact us at [email protected] for pricing (be sure to provide your manuscript number and title). Alternatively, you should make your own arrangements to improve the language quality and provide details in your response letter. – PeerJ Staff

·

Basic reporting

This study presents a sentiment generation model tailored for teacher-student course interactions. The model encompasses data preprocessing and enhancement techniques, a multimodal sentiment recognition model for dialogue texts, and a topic-extended sentiment dialogue generation model Although the results of this paper have excellent value, there are also some problems, some revisions needed to be revised to make sure that the manuscript can be accepted. The commonly problems are as follows:

1. You highlighted the problem of limited interaction in online education compared to traditional methods. However, it would be beneficial to include a brief statistical reference or source to emphasize the significance of this issue.

2. The abstract should mention what kind of limitations the existing research on emotion generation models faces in more concrete terms. Specify examples of these limitations, if possible.

Experimental design

3. It's important to clearly define what you mean by "emotional communication" in the context of online physical education teaching to ensure readers understand the scope of your study.

4. Provide a concise summary of the key findings from the simulation experiments, including any insights gained from comparing different parameter settings.

5. Consider addressing the potential future implications and the broader relevance of your study in the field of education.

Validity of the findings

6. Consider providing a brief statement on the potential implications or benefits of improving emotional communication in online physical education teaching at the end of this section.

Additional comments

7. Use more descriptive language to convey the innovative aspect of your proposed emotion generation model, as this will help engage readers from the beginning.

8. There are also some problems in language expression in this paper, which need to be modified. Please check the Chinese characters in the replacement formula and the redundant space characters in the references.

Reviewer 2 ·

Basic reporting

I have had the opportunity to review the paper. Your research on emotion generation models in the context of teacher-student course interactions is certainly intriguing and relevant. The following inputs might help the author/s to improve the paper. Authors should modify the article following the comments indicated below to increase the quality of research justification, contributions, originality and findings.

 Begin the second part by briefly summarizing the challenges faced by sentiment generation models in the context of teacher-student course interactions to provide context for your proposed solution.
 Clearly explain the ECM model, including its acronym, before introducing the LSTM model as a solution. Define the terminology to ensure clarity.
 It would be beneficial to mention the sources of data used for your simulation experiments, as this will add credibility to your results.
 Consider elaborating on the significance of the "contextual information" in sentiment generation. How does it affect the model's performance?
 In the concluding sentences, reiterate the practical applications of your approach in online physical education and how it can enhance teaching quality and learning effectiveness.
 Ensure a smooth transition from discussing the proposed model to presenting the experimental results, maintaining a clear flow of ideas.
 In your concluding statement, express your confidence in the study's contribution to the field and its potential to advance emotional communication and learning dynamics in online physical education.
 Avoid repeating phrases or ideas unnecessarily. For example, try to rephrase the use of "sentiment generation" and "emotional communication" to keep the language fresh.
 Lastly, remember to maintain a consistent tone and style throughout the abstract to enhance readability and cohesiveness.

Experimental design

.

Validity of the findings

.

---

## Round 0.2 · accepted · Accept

Dear authors,

We have received the reviewers' reports on your manuscript. Thank you for addressing the concerns, questions and suggestions on the review reports. The paper seems to have improved in the opinion of the reviewers. At the moment your article seems to be acceptable for publication after the last revision.

Best wishes,

·

Basic reporting

no comment

Experimental design

no comment

Validity of the findings

no comment

Additional comments

no comment

Reviewer 2 ·

Basic reporting

Please the authors have addressed the changes carefully in revised version and it’s suitable in its current form

Experimental design

Please the authors have addressed the changes carefully in revised version and it’s suitable in its current form

Validity of the findings

Please the authors have addressed the changes carefully in revised version and it’s suitable in its current form